# Comparison between Measured and Predicted Resting Metabolic Rate Equations in Cross-Training Practitioners

**DOI:** 10.3390/ijerph21070891

**Published:** 2024-07-09

**Authors:** Ana Flávia Sordi, Bruno Ferrari Silva, Breno Gabriel da Silva, Déborah Cristina de Souza Marques, Isabela Mariano Ramos, Maria Luiza Amaro Camilo, Jorge Mota, Pablo Valdés-Badilla, Sidney Barnabé Peres, Braulio Henrique Magnani Branco

**Affiliations:** 1Interdisciplinary Laboratory of Intervention in Health Promotion, Cesumar University, Maringá 87050-390, Paraná, Brazil; anaflaviasordi@gmail.com (A.F.S.); brunoferrarisilva@live.com (B.F.S.); marques.deborah@hotmail.com (D.C.d.S.M.); isabelaramos94@gmail.com (I.M.R.); malucamilofisio@gmail.com (M.L.A.C.); 2Department of Physiological Sciences, State University of Maringá, Maringá 87020-900, Paraná, Brazil; sbperes@uem.br; 3Luiz de Queiroz College of Agriculture–ESALQ, USP Department of Exact Sciences, University of Sao Paulo, Sao Paulo 13418-900, Sao Paulo, Brazil; brenogsilva@usp.br; 4Graduate Program in Health Promotion, Cesumar University, Maringá 87050-390, Paraná, Brazil; 5Research Center in Physical Activity, Health and Leisure (CIAFEL), Faculty of Sports, University of Porto (FADEUP), Porto 4200-450, Portugal; jmota@fade.up.pt; 6Laboratory for Integrative and Translational Research in Population Health (ITR), Porto 4050-600, Portugal; 7Department of Physical Activity Sciences, Faculty of Education Sciences, Universidad Católica del Maule, Talca 3530000, Chile; valdesbadilla@gmail.com; 8Sports Coach Career, School of Education, Universidad Viña del Mar, Viña del Mar 2520000, Chile

**Keywords:** athletic, energy expenditure, calorimetry, extreme functional fitness training

## Abstract

This study aimed to investigate the resting metabolic rate (RMR) in cross-training practitioners (advanced and novice) using indirect calorimetry (IC) and compare it with predictive equations proposed in the scientific literature. Methods: A cross-sectional and comparative study analyzed 65 volunteers, both sexes, practicing cross-training (CT). Anthropometry and body composition were assessed, and RMR was measured by IC (FitMate PRO^®^), bioimpedance (BIA-InBody 570^®^), and six predictive equations. Data normality was tested by the Kolgomorov–Smirnov test and expressed as mean ± standard deviation with 95% confidence intervals (CI), chi-square test was performed to verify ergogenic resources, and a Bland–Altman plot (B&A) was made to quantify the agreement between two quantitative measurements. One-way ANOVA was applied to body composition parameters, two-way ANOVA with Bonferroni post hoc was used to compare the RMR between groups, and two-way ANCOVA was used to analyze the adjusted RMR for body and skeletal muscle mass. The effect size was determined using Cohen’s d considering the values adjusted by ANCOVA. If a statistical difference was found, post hoc Bonferroni was applied. The significance level was *p* < 0.05 for all tests. Results: The main results indicated that men showed a higher RMR than women, and the most discrepant equations were Cunningham, Tinsley (b), and Johnstone compared to IC. Tinsley’s (a) equation indicated greater precision in measuring the RMR in CM overestimated it by only 1.9%, and BIA and the Harris–Benedict in CW overestimated RMR by only 0.1% and 3.4%, respectively. Conclusions: The BIA and Harris–Benedict equation could be used reliably to measure the RMR of females, while Tinsley (a) is the most reliable method to measure the RMR of males when measuring with IC is unavailable. By knowing which RMR equations are closest to the gold standard, these professionals can prescribe a more assertive diet, training, or ergogenic resources. An assertive prescription increases performance and can reduce possible deleterious effects, maximizing physical sports performance.

## 1. Introduction

Cross training (CT) is a strength and conditioning exercise program that uses different physical capacities, i.e., cardiorespiratory fitness, maximal strength, muscle power, speed, agility, muscle endurance, balance, and flexibility, as well as energy systems, i.e., anaerobic lactic, alactic and oxidative along of different specifics workout for each day (workout of the Day—WOD) [1]. CT sessions comprise a range of functional movements involving the whole body [1,2,3,4], including calisthenic activities, strength and power, weightlifting, gymnastic movements, plyometric exercises, cycling, running, and rowing, which can be performed at high-intensity [2,3,4]. Thus, CT is an excellent form of physical exercise for healthy adults since it increases maximum oxygen consumption (VO_2_ max), maximum strength, muscle hypertrophy, and muscle endurance [5,6,7]. All these responses are like those found in other high-intensity modalities and may increase resting metabolic rate (RMR) [6,7].

The RMR is the body’s minimum energy to maintain vital functions under basal conditions [8]. Sufficient energy is critical for training consistency since prolonged energy restriction can impair physiological function, increasing the risk of injuries and fatigue [8]. Considering that low energy availability can harm performance, the accuracy of RMR measurements becomes essential to monitoring that energy [8]. The RMR represents approximately 60–70% of total energy expenditure in sedentary individuals and up to 50% in athletes [9]. So, the assertive estimation of RMR is crucial to establishing the nutritional strategy for the athlete according to body needs and avoiding conditions of fatigue and muscle mass loss [8].

With the advent of technology, different methods have been developed to assess the RMR [9,10]. The gold standard for measurement of RMR is performed via indirect calorimetry (IC), a technique that measures gas exchange, i.e., oxygen consumption (VO_2_) and carbon dioxide (CO_2_) production, and the interrelation between them is called respiratory quotient (RQ) [11]. These processes are associated with the oxidation of the leading energy substrates, such as carbohydrates and lipids, allowing energy expenditure estimation [9]. Although IC is highly accurate, the high cost of the equipment prevents its large-scale use [12]. Thus, to minimize costs and improve evaluation methods, other more accessible assessment parameters were proposed in the scientific literature to optimize training models [13]. As a result, differing predictive equations were developed and validated from the gold standard measurement for general and specific populations [14].

The RMR is recognized as one of the main determinants of an athlete’s energy needs according to the specificity of the modality [10]. However, specific equations for CT practitioners (novice and advanced) have not yet been published in scientific databases based on the author’s knowledge. The predictive equations were developed for specific populations and may erroneously estimate the RMR when used in populations different from those proposed by the offer, thus limiting their applicability to athletes, specifically in the modality in question. Previous studies have analyzed the validity of predictive equations in athletes from different sports. In bodybuilders, the De Lorenzo method proved to be adequate to measure the RMR of women, and the Tinsley equation proved to be the most appropriate method to quantify the RMR of men [10]. In addition, bioelectrical impedance analysis (BIA) (InBody 570^®^), Harris–Benedict, and Cunningham underestimated the RMR of bodybuilding athletes. In college students, BIA (InBody 570^®^) underestimated the RMR [9]. On the other hand, for football, track and field, swimming, and baseball athletes, the Harris–Benedict equation was shown to predict RMR values more accurately in men and the Cunningham equation in women [15]. When evaluating RMR in rowers and canoeists, the Cunningham and Harris–Benedict equations were underestimated in male athletes but not in female athletes [16].

These findings demonstrate that the studies have been based on estimating the RMR from the predictive equations and comparing them with the result of the RMR measured in different populations [17,18]. In different studies, biological and behavioral characteristics have significantly interfered with both methods, predicted and measured [19,20]. Thus, considering CT’s growing popularity and competition, ref. [21], measuring the RMR of CT practitioners, could contribute satisfactorily to the direction of nutritional behavior and possible improvement of the physical sports performance of these sportsmen. Therefore, the main aim of this study was to investigate the RMR in cross-training practitioners (advanced and novice) using indirect calorimetry (IC) and compare it with predictive equations proposed in the scientific literature.

## 2. Materials and Methods

### 2.1. Study Design

The present study was a cross-sectional and comparative design, comprising 65 participants of both sexes, with advanced and novice CT practitioners. A probabilistic sample based on a previous study with the same methodology indicated that 61 participants were enough to identify an *α* = 0.05 and *β* = 0.80 [10]. The volunteers were initially divided into four groups: novice men practicing CT (NM, *n* = 15; 29.7 ± 6.0 years old), novice women practicing CT (NW, *n* = 17; 29 ± 5.6 years old), advanced men practicing CT (AM, *n* = 16; 28.5 ± 5.3 years old) and advanced women practicing CT (AW, *n* = 17; 30.0 ± 5.5 years old).

Data were collected at the university campus at approximately 8 a.m., by prior scheduling, in the following order: (i) anamnesis to collect information on the participant’s medical history, nutrition profile, and use of anabolic steroids, routine, training, and last competition; (ii) blood pressure measurement at rest; (iii) height measurement; (iv) body composition analysis; and (v) assessment of RMR with IC. RMR values were calculated using predictive equations (which are presented in the sections below).

The guidelines for the procedures were as follows: (i) fast for 8 h, that is, do not ingest solids or liquids (including water), (ii) avoid using any diuretic substances for at least 24 h prior to the procedures, (iii) interrupt moderate- or high-intensity exercises on the day before the test, (iv) wear light clothes, (v) urinate or evacuate about 30’ before the tests, (vi) do not use metallic objects, and (vii) do not consume caffeine-based drinks 12 h prior to the test [22,23]. Also, the participants were instructed to maintain their feeding routine for 24 h before the assessment. According to the specifications of a previous study [24], the laboratory temperature was maintained at 24 °C.

All participants of the study signed a consent form. The research was conducted at the University’s Exercise Physiology Laboratory at Cesumar University (Maringa–Brazil). The Ethics and Local Research Committee approved the study (protocol no. 4,546,726), following the recommendations proposed by resolution 466/12 of the Brazilian Ministry of Health and the Declaration of Helsinki.

### 2.2. Participants

Participants were allocated into different groups after the interview, using the following inclusion criteria: (i) CT regularly practiced for at least 6 months (novice and advanced); (ii) advanced practitioners self-identifying as one and having competed in the last two years; and (iii) training frequency > 3 sessions a week for participants of the two groups of practitioners (novice and advanced). All participants were asked about the use of ergogenic aids and anabolic steroids. The volunteers who presented conditions that could reduce energy expenditure (osteomyoarticular injuries, chronic diseases, or physical limitations), practiced other modalities, or did not follow the previously requested protocol were excluded.

The intentional allocation of the groups was organized by physical training history and participation in CT competitions; that is, participants that practiced CT regularly, for health or fitness purposes, for at least 6 months with training frequency greater than 3 sessions a week and did not participate in CT competitions in the last 2 years were allocated to the novice women CT (NW) and novice men CT (NM) groups, while the participants that practiced CT regularly, for performance, for at least 6 months, with training frequency greater than 3 sessions a week and self-identification as athletes, having competed in the last two years were allocated in advanced women CT (AW) and advanced men CT (AM) groups.

### 2.3. Anthropometry and Body Composition

Height was measured following the procedure proposed by Lohman, Roche, and Martorell [25] through a stadiometer (Sanny, standard model, São Paulo, Brazil). Body composition was assessed via InBody 570^®^–BIA (Bio space Co., Ltd., Seoul, Republic of Korea) (BIA). BIA is considered a non-invasive, practical, and double indirect measurement that is used to calculate resistance and electrical reactance through electrical passage [9], thus evaluating the following variables: (i) body mass; (ii) lean mass; (iii) skeletal muscle mass (SSM); (iv) fat mass; (v) body fat percentage; and (vi) RMR.

### 2.4. Measurement of Resting Metabolic Rate via Indirect Calorimetry

RMR was measured using a Fitmate metabolic gas analyzer (model PRO^®^, COSMED, Rome, Italy). According to the manufacturer’s manual, the equipment was self-calibrated before each analysis. All participants were previously instructed to perform the measurement. The volunteers wore silicone masks fixed on their faces and remained supine for 15 min and 15 s [10]. The FitMate PRO^®^ is a standard metabolic analyzer that calculates energy expenditure using a fixed respiratory quotient (VCO_2_/VO_2_) of 0.85 [9]. The values calculated in Fitmate PRO^®^ are used in the simplified equation proposed by Weir [26].

### 2.5. Predictive Equations for Calculating Resting Metabolic Rate

The data obtained by the BIA test were used to calculate the RMR of all volunteers using the following predictive equations: Harris and Benedict [27], Cunningham [13], De Lorenzo [28], Tinsley [17], Johnstone [29], and BIA InBody 570^®^ [30], which can be based on body mass, height, age, skeletal muscle mass, and fat mass according to the specificity of the equation (Table 1). Harris and Benedict [27], Cunningham [13], and De Lorenzo [28] were chosen because of their affinity and predictive effect on athletic populations [6] or people in good health; Johnstone [29] was included as it sought to improve Schofield’s (1985) method; and Tinsley’s equations (a) and (b) were included since they are newly developed equations for bodybuilders [17]. Most equations use kilocalories (kcal) as a unit of measurement, except for the Johnstone equation, which is expressed in kilojoules (kJ). Thus, kilojoules were converted to kilocalories to standardize the measurement units [31].

### 2.6. Statistical Analysis

Data normality was tested by the Kolgomorov–Smirnov test, and after the respective confirmation of normality (*p* > 0.05), data were expressed as mean, ± standard deviation, and 95% confidence intervals (CI). An analysis of variance (one-way ANOVA) with Bonferroni post hoc was applied to identify possible differences in body composition parameters between the groups (AM vs. NM and AW vs. NW). A chi-square test was performed to verify the frequency distribution data for using ergogenic resources. A Two-way ANOVA with Bonferroni post hoc was applied to compare the RMR (among gold standard measurement, BIA, and different predictive equations). Moreover, a two-way with Bonferroni post hoc covariance (ANCOVA) was used to analyze the RMR adjusted for body and skeletal muscle mass. The Bland–Altman plot (B&A) was conducted to quantify the agreement between two quantitative measurements. According to the recommendation, the difference between the methods was calculated using the following equation: [(method A − method B)/mean of both methods × 100] [32]. The effect size was determined using Cohen’s d, categorized as follows: small effect (0.2), moderate effect (0.5) and large effect (0.8), considering the values adjusted by ANCOVA [33]. The significance level was *p* < 0.05 for all statistical tests performed. Statistical analysis was performed using SPSS version 22.0 (IBM, Armonk, NY, USA).

## 3. Results

Table 2 presents advanced and novice practitioners’ anthropometric and body composition characteristics, comparing the groups between sex (NW vs. NM and AW vs. AM) and novice versus same-sex advanced practitioners (NW vs. AW and NM vs. AM). The results indicated that both groups, AM and NM, presented higher values for height, body mass, lean mass, skeletal muscle mass, and RMR (IC measurement) and lower values for body fat percentage (*p* < 0.05) when compared to AW and NW. No significant differences were detected for any variable for advanced or novice practitioners in sex comparisons (*p* > 0.05).

Table 3 shows the nutritional profile and use of anabolic steroids in the groups evaluated. Based on the results, none of the groups presented a significant difference in the consumption of energy supplements (X^2^ = 1.14; p = 0.7), vitamins (X^2^ = 2.01; *p* = 0.5), and anabolic steroids (X^2^ = 3.21; *p* = 0.3). Thus, it can be observed that there was no association between the consumption of ergogenic resources with any of the groups evaluated (*p* > 0.05).

Table 4 presents RMR values in IC, BIA, and predictive equations. When comparing the results of RMR in women and men (novice and advanced), no significant differences were found between the same sexes in different groups (NW vs. AW and NM vs. AM; (*p* > 0.05 for all comparisons). On the other hand, when comparing the sexes (NW vs. NM and AW vs. AM), it was observed that significant differences were found in all methods (*p* < 0.05). Based on the previous results, the analysis of agreement among the IC, BIA, and predictive equations, as well as the difference between the methods and covariance, were performed by analyzing the same-sex groups together (CT men and CT women), given the absence of differences between the novices and advanced in the RMR estimated in all methods (*p* > 0.05).

Figure 1 shows the plots of the RMR of CT men (CM). The CM values were as follows: (A) IC and BIA: BIAS = −169.6, SD = 344.7, CI upper = −845.2, CI lower = 506.0; (B) IC and Harris and Benedict: BIAS = 149.4, SD = 349.0, CI upper = −534.8, CI lower = 833.5; (C) IC and Cunningham: BIAS = 688.6, SD = 355.1, CI upper = −7.325, CI lower = 1385; (D) IC and De Lorenzo: BIAS = 96.6, SD = 351.4, CI upper = −592.2, CI lower = 785.5; (E) IC and Tinsley (a): BIAS = −66.7, SD = 355.7, CI upper = −763.9, CI lower = 630.4; (F) IC and Tinsley (b): BIAS = 741.6, SD = 350.9, CI upper = 53.8, CI lower = 1429; (G) IC and Johnstone: BIAS = 797.8, SD = 348.3, CI upper = 115.2, CI lower = 1480.

Figure 2 shows the plots of the RMR of CT women (CW). The CW responses indicated that (A) IC and BIA: BIAS = −88.0, SD = 189.1, CI upper = −282.5, CI lower = 458.6; (B) IC and Harris and Benedict: BIAS = −253.7, SD = 192.5, CI upper = −630.9, CI lower = 123.6; (C) IC and Cunningham: BIAS = 215.4, SD = 190.5, CI upper = −157.9, CI lower = 588.8; (D) IC and De Lorenzo: BIAS = −342.0, SD = 208.8, CI upper = −751.2, CI lower = 67.14; (E) IC and Tinsley (a): BIAS = −248.2, SD = 235.6, CI upper = −709.9, CI lower = 213.6; (F) IC and Tinsley (b): BIAS = 332.3, SD = 189.3, CI upper = −38.8, CI lower = 703.4; (G) IC and Johnstone: BIAS = 290.9, SD =213.6, CI upper = −127.8, CI lower = 709.5.

The mean percentage differences among the methods for men about IC were as follows: BIA estimated for less in 5.8%; Harris–Benedict estimated for less in 4.9%; Cunningham estimated for less in 24.5%; De Lorenzo estimated for less in 3.2%; Tinsley (a) estimated for more at 1.9%; Tinsley (b) estimated for less at 26.6%; Johnstone estimated for less at 28.9%. For women, the mean percentage differences found in the methods with IC were as follows: BIA estimated for more in 0.1%; Harris-Benedict estimated for more in 3.4%; Cunningham estimated for less in 16.2%; De Lorenzo estimated for more in 9.9%; Tinsley (a) estimated for more at 6.9%; Tinsley (b) estimated for less at 22.6%; Johnstone estimated for less at 22.3%.

When adjusting the RMR analysis by the covariate body mass (Table 5), it was observed that in CW, the equations of Cunningham, Tinsley (b), and Johnstone showed significant differences underestimating the RMR in 17.2% (*p* = 0.000, d = 9.77; large effect), 48.5% (*p* = 0.000, d = 13.12; large effect), and 23.6% (*p* = 0.000, d = 8.67; large effect), respectively, while De Lorenzo and Tinsley (a) showed significant differences overestimating the RMR in 10.7% (*p* = 0.000, d = 6.93; large effect) and 7.8% (*p* = 0.000, d = 5.01; large effect), respectively, to the gold standard (IC) (*p* < 0.05); the CM group showed significant differences between the means for BIA, underestimating the RMR in 5.4% (*p* = 0.000, d = 3.52; large effect), Harris–Benedict underestimating the RMR in 4.6% (*p* = 0.000, d = 3.05, large effect), Cunningham underestimating RMR by 23.3% (*p* = 0.000, d = 14.10; large effect), De Lorenzo underestimating RMR by 3.2% (*p* = 0.05, d = 2.12; moderate effect), Tinsley (b) underestimating RMR by 50.3% (*p* = 0.000, d = 15.13; large effect), and Johnstone underestimating RMR by 27.6% (*p* = 0.000, d = 16.21; large effect) compared to the gold standard (IC) (*p* < 0.05).

When adjusting the RMR analysis by the covariate skeletal muscle mass (Table 6), it was observed that in CW, the equations of Cunningham, Tinsley (b), and Johnstone showed significant differences in estimating, to a lesser extent, the RMR in 17.9% (*p* = 0.000, d = 6.66; large effect), 49.5% (*p* = 0.000, d = 10.47; large effect), and 24.4% (*p* = 0.000, d = 9.48; large effect), respectively, while De Lorenzo and Tinsley (a) overestimated the RMR at 10.3% (*p* = 0.000, d = 8.09; large effect) and 7.4% (*p* = 0.000, d = 4.93; large effect), respectively, with the IC (*p* < 0.05); the CM group showed significant differences between the means for BIA, underestimating RMR in 5.2% (*p* = 0.000, d = 4.53; large effect), Harris–Benedict underestimating RMR in 4.6% (*p* = 0.000, d = 4.46; large effect), Cunningham underestimating RMR in 23% (*p* = 0.000, d = 10.42; large effect), De Lorenzo underestimating RMR in 3.2% (*p* = 0.05, d = 8.07; moderate effect), Tinsley (b) underestimating RMR in 49.8% (*p* = 0.000; d = 13.89, large effect), and Johnstone underestimating RMR in 27% (*p* = 0.000, d = 12.17; large effect) compared to the gold standard (IC) (*p* < 0.05).

## 4. Discussion

The present study has two aims: 1. to investigate the RMR in cross-training practitioners (advanced and novice) using indirect calorimetry (IC), and 2. to compare it with predictive equations proposed in the scientific literature. In summary, the main findings were as follows: (i) men (advanced and novices) had a higher RMR when compared to women (AM vs. AW; NM vs. NW) (Table 2); (ii) the difference between the methods demonstrated that the most discrepant equations were Cunningham, Tinsley (b), and Johnstone when compared to IC in CW and CM; (iii) the equations that indicated less variation according to B&A and percentage difference were De Lorenzo and Tinsley (a) for men (CM) (Figure 1), and BIA and Harris–Benedict for women (CW) (Figure 2); (iv) with the adjustment of RMR for the covariates body mass and skeletal muscle mass, the same response was observed, that is, the BIA and Harris–Benedict equations approached the gold standard in females (CW), while only the Tinsley (a) equation approached the gold standard in males (CM) (Table 5 and Table 6).

Anthropometric variables showed significant differences in height, body mass, lean mass, skeletal muscle mass, body fat percentage, and RMR when comparing novice and advanced male practitioners with the women novice and advanced practitioners. It is known that physiological differences based on sex affect body composition and sports performance. Men have high testosterone levels that stimulate skeletal muscle hypertrophy and type II muscle fiber synthesis, giving them a natural athletic advantage over women [34,35,36]. However, the body composition variables did not show differences when comparing practitioners of the same sex, that is, NM vs. AM and NW vs. AW, unlike previous studies that observed that athletes (advanced practitioners) have a lower percentage of body fat and higher fat-free mass when compared to novice practitioners, indicating differences in body composition [37]. About RMR, no differences were found when comparing novices and advances of the same sex, corroborating previous evidence [37] and showing that both the practice of CT aiming at performance and recreational practice does not change the RMR.

RMR can be affected by the body composition of an athlete, as well as by dietary and nutritional habits, influencing athletic performance [38]. However, the frequency distribution data showed no differences between the consumption of energy supplements, vitamins, and anabolic steroids among novices and advanced practitioners of CT, indicating a homogeneity between the groups in the nutritional profile. These results do not corroborate previous findings that showed differences in self-reported rates for the consumption of energy supplements, vitamins, and anabolic steroids by athletes, particularly bodybuilding athletes, which justifies the increase in RMR in bodybuilders who use this type of ergogenic resources when compared to bodybuilders [10].

Considering the absence of differences in RMR estimated in all methods between the groups, novice vs. advanced practitioners of the same sex (AM vs. NM; AW vs. NW), the groups were relocated to CM (CT men) and CW (CT women) for the subsequent analysis. The measurement methods (IC and BIA) and measurement (predictive equations) consider different variables to quantify the RMR. Thus, B&A plots, followed by the Giavarina protocol, are recommended to identify the difference between the methods [32]. In addition, considering that previous research supports the validity and reliability of IC [22], the present study adopted the RMR values measured by FitMate PRO^®^ as the gold standard.

The B&A plots of IC, BIA, and predictive equations indicated low variability for the De Lorenzo equation, underestimating by 3.2%, and Tinsley (a), overestimating by 1.9% in CM, while BIA and Harris–Benedict overestimated by 0.1% and 3.4%, respectively, indicating good reliability and greater precision of these equations when compared to the gold standard method. On the other hand, the Cunningham and Tinsley (b) equations underestimated RMR by 24.5% and 26.6%, respectively, when compared to IC in men, and the Cunningham and Tinsley (b) equations underestimated RMR at 16.2% and 22.6%, respectively, in women. The Johnstone equation underestimated RMR at 28.9% in men and 22.3% in women, indicating low reliability for CW and CM. Their results indicate that the Cunningham, Tinsley (b), and Johnstone methods are unsuitable for CM and CW (novice and advanced).

The RMR was adjusted for body and skeletal muscle mass variables to confirm the present finding. The results of the present study suggest that when adjusting the RMR for the variables body mass and skeletal muscle mass, the De Lorenzo method differs statistically from the gold standard (IC) method, underestimating the RMR in CM (novice and advanced).

Therefore, if the variables body mass and skeletal muscle mass are essential for male CT practitioners (novice and advanced) to achieve their goal, such as reducing body mass, weight loss, performance, and increasing skeletal muscle mass, the use of the De Lorenzo equation is not recommended. Alternatively, the Tinsley equation (a), which did not present a significant difference compared to the IC in both variables, can be adopted.

In relation to women, when adjusting for the covariates body mass and skeletal muscle mass, a similar response was observed, confirming the reliability and applicability of BIA and the Harris–Benedict method in predicting RMR in CW (novice and advanced). These results do not corroborate the previous finding that analyzed the applicability of the equation to bodybuilding athletes and realized that the De Lorenzo method could be used reliably to measure the RMR of female athletes [10].

It is noteworthy that the Tinsley (a) and Harris–Benedict equations, which indicated good reliability and greater precision in CM and CW, respectively, use body mass to quantify RMR in their equations, showing the importance of this variable in the accurate and reliable measurement of RMR [29] and corroborating with previous findings that indicate that the methods based on body mass minimize the chances of underestimating or overestimating the RMR, being the most practical option to quantify RMR [17,39].

On the other hand, the equations that indicated greater discrepancy in the quantification of RMR, that is, Cunningham, Tinsley (b), and Johnstone, use fat-free mass or skeletal muscle mass as a variable in their equations, not corroborating with previous findings that indicated that the Cunningham equation predict more accurately the RMR in men and women endurance-trained [40] and recreational athletes [41]. The high Cohen’s d values observed in our study indicate that the RMR measurement methods (indirect calorimetry vs. predictive equations) produce significantly different results. This reflects the varying sensitivity and accuracy of the methods.

The outcomes of the present study indicated that the Tinsley equation (a), developed specifically for bodybuilding athletes [10], close to the IC in the adjusted analysis, B&A, and mean percentage difference, can be applied to measure the RMR in men practicing CT, and regarding women practicing CT, BIA can be used, while the Harris–Benedict equation presented values close to those obtained with IC in all analyses, indicating the accuracy and reliability in their equations.

The present study’s findings reinforce the premise that the predictive equations were developed for specific audiences and that they are not reliable and valid when applied in groups of individuals with distinct metabolic conditions [42] and highlight the importance of considering the specificity of the modality in question.

Thus, before applying a predictive equation to quantify the RMR in each group, one must determine which is the most appropriate for the public [8], considering the sport, nationalities, and body composition variables. For example, if body mass is a key factor that can impact sports performance, as in the case of martial arts athletes [43], one could consider using predictive methods that use body mass to quantify the RMR.

Therefore, considering that providing sufficient energy is extremely important for improving sports performance and physiological and metabolic aspects [44] and that low energy availability negatively impacts the health of athletes [45], professionals working with sports nutrition, sports medicine, and sports training may use the information provided to improve the sports performance of practitioners, providing accurate measurement of the RMR according to the specificity of the modality in question.

From this perspective, the present study’s findings have important practical implications for professionals who work with CT practitioners. The RMR will be estimated more precisely using the most appropriate predictive equation, adjusting the eating plan and training according to the practitioner’s objectives, thus optimizing performance and recovery [8]. For nutritionists, understanding RMR with greater accuracy means providing dietary recommendations that better sustain energy demands. Conversely, coaches can use these findings to plan training sessions and adjust intensity, volume, and energy expenditure.

The present study has some limitations. The sample are homogeneous; therefore, the findings should not be extrapolated and applied to populations with different conditions. The IC adopted as the gold standard is a measurement method based on gas exchange, while indirect calorimetry is a more expensive method that controls more variables, such as gases, temperature, and humidity, offering greater accuracy in the results. Furthermore, the period of training, which varies significantly between practitioners, was not considered, which could influence RMR; this means the study was limited to including practitioners inserted in the competitive environment, regardless of the training periodization and competitive level. Finally, dietary records and information about the participants’ sleeping patterns were not collected. The absence of these data may impact the results, as they are factors that influence metabolism.

Future research could focus on analyzing a more representative, diverse and heterogeneous sample, including practitioners of different ages, nutritional profile, fitness levels, and periodization levels as well as longitudinal research examining the impact on RMR. Furthermore, future research can focus on developing a predictive equation for advanced and novice CT practitioners.

No previous study has evaluated the RMR of CT practitioners (advanced and novice) and compared it with the literature’s proposed predictive equations. Thereby, this study is a valuable tool for sports nutrition, sports medicine, and sports training, who can use this article’s information to increase the sports performance of practitioners of CT.

## 5. Conclusions

In conclusion, BIA and the Harris–Benedict equation could be used reliably to measure the RMR of females, while the Tinsley (a) equation is the most reliable method to measure the RMR of males when measuring with IC is unavailable. By knowing which RMR equations are closest to the gold standard, these professionals can prescribe a more assertive diet, training, or ergogenic resources. An assertive prescription increases performance and can reduce possible deleterious effects, maximizing physical sports performance. The other equations should be used cautiously when applied to CF practitioners, especially Cunningham, Tinsley (b), and Johnstone.

## Figures and Tables

**Figure 1 ijerph-21-00891-f001:**
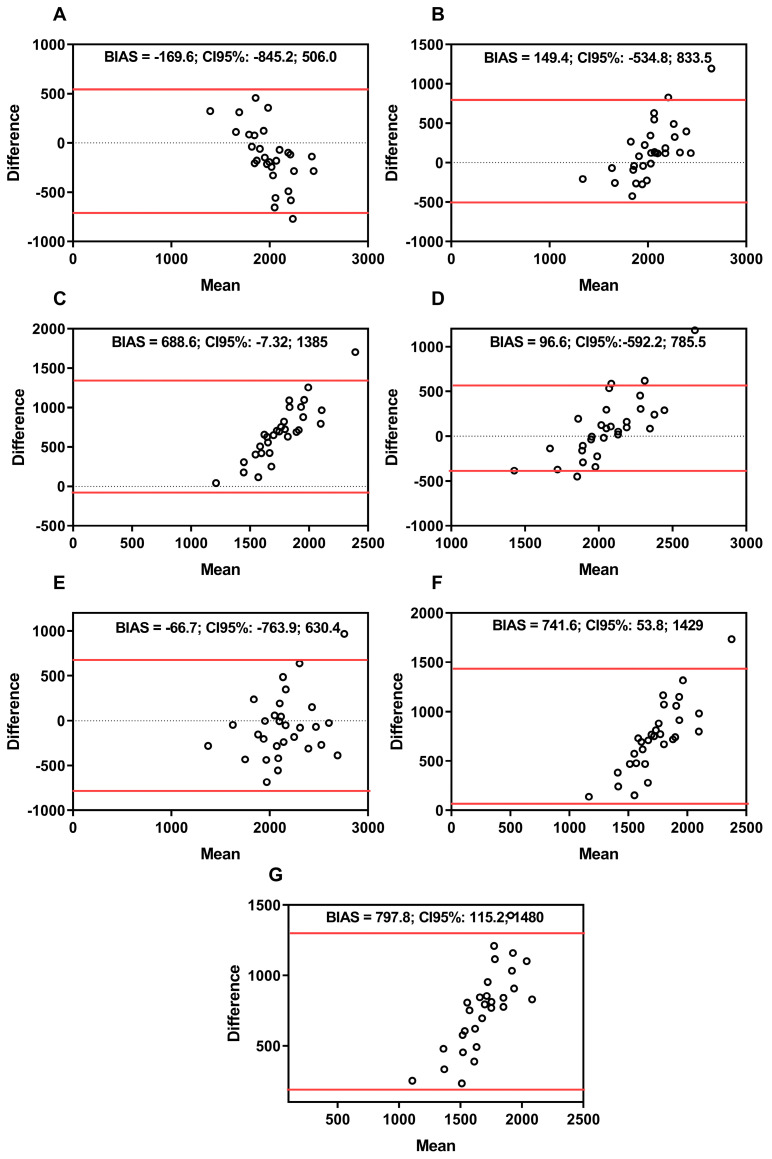
Bland–Altman plots between indirect calorimetry, bioelectrical impedance, and predictive equations of men (advanced and novices). Note: panel (**A**) IC and BIA; panel (**B**) IC and Harris and Benedict equation; panel (**C**) IC and Cunningham equation; panel (**D**) IC and De Lorenzo equation; panel (**E**) IC and Tinsley (a) equation; panel (**F**) IC and Tinsley (b) equation; panel (**G**) IC and Johnstone equation. Black line = mean; red line = 95% confidence interval (upper and lower). BIAS = difference between the measurements. Kcals = kilocalories.

**Figure 2 ijerph-21-00891-f002:**
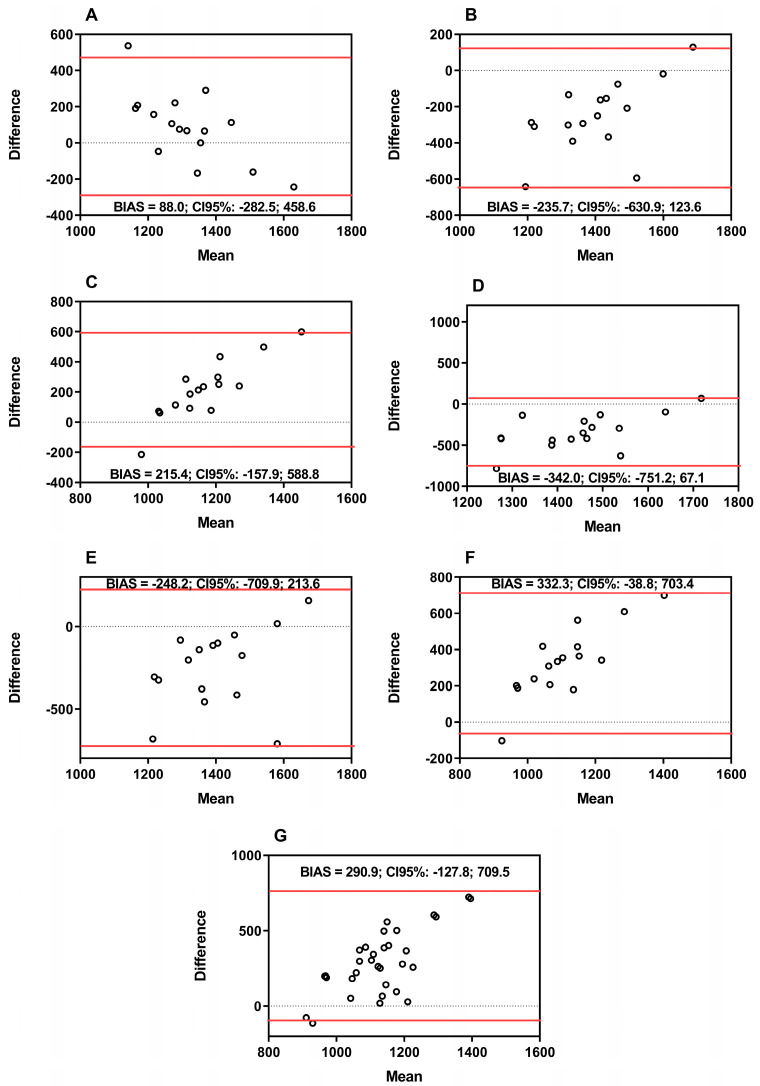
Bland Bland–Altman plots between indirect calorimetry, bioelectrical impedance, and predictive equations of women (advanced and novices). Note: panel (**A**) IC and BIA; panel (**B**) IC and Harris and Benedict equation; panel (**C**) IC and Cunningham equation; panel (**D**) IC and De Lorenzo equation; panel (**E**) IC and Tinsley (a) equation; panel (**F**) IC and Tinsley (b) equation; panel (**G**) IC and Johnstone equation. Black line = mean; red line = 95% confidence interval (upper and lower); BIAS = difference between the measurements. Kcals = kilocalories.

**Table 1 ijerph-21-00891-t001:** Resting metabolic rate prediction equations.

Reference	Equation
Harris and Benedict (1918) [27]	Men RMR (kcal/d) = 66.47 + 13.75 × BM + 5 × Height − 6.76 × Age Women RMR (kcal/d) = 655.7 + 9.56 × BM + 1.85 × Height − 4.68 × Age
Cunningham (1991) [13]	RMR (kcal/d) = 500 + 22 × FFM
De Lorenzo (1999) [28]	RMR (kcal/d) = −857 + 9 × BM + 11.7 × Height
Tinsley (a) (2018) [17]	RMR (kcal/d) = 24.8 × BM + 10
Tinsley (b) (2018) [17]	RMR (kcal/d) = 25.9 × FFM + 284
Johnstone (2016) [29]	RMR (kJ/d) = 90.2 × FFM + 31.6 × FM − 122 × Age + 1613
BIA InBody 570^®^ [30]	RMR (kcal/d) = 21.6 × LM + 370

Note: RMR = resting metabolic rate; BM = body mass; FFM = fat-free mass; FM = fat mass; LM = lean mass units for equations: BW (kg); height (cm); age (years); FFM (kg); FM (kg).

**Table 2 ijerph-21-00891-t002:** Age, anthropometric parameters, body composition, and resting metabolic rate of cross-training practitioners.

Variables	NW (*n* = 17)	NM (*n* = 15)	AW (*n* = 17)	AM (*n* = 16)
Age (years old)	29.5 ± 5.4	29.7 ± 6.0	30.0 ± 5.5	28.5 ± 5.3
Height (cm)	164.5 ± 5.4	180.4 ± 5.4 #	164.1 ± 7.2	175.2 ± 5.8 *
Body mass (kg)	61.1 ± 5.6	88.5 ± 10.9 #	63.6 ± 8.8	85.8 ± 13.1 *
Lean mass (kg)	43.2 ± 3.9	68.3 ± 6.5 #	47.3 ± 6.9	68.4 ± 9.0 *
Fat mass (kg)	15.1 ± 4.2	15.8 ± 6.8	13.3 ± 3.2	13.3 ± 5.3
Skeletal muscle mass (kg)	25.4 ± 2.52	41.5 ± 3.9 #	28.0 ± 4.4	41.8 ± 5.9 *
Body fat percentage (%)	24.6 ± 5.3	17.6 ± 5.9 #	21.0 ± 3.8	15.7 ± 5.5 *
RMR (kcal)	1275 ± 209.5	2147 ± 320.0 #	1530 ± 375.8	2069 ± 469.9 *

Note: data are presented as mean ± standard deviation; RMR = resting metabolic rate; NW = novice women; NM = novice men; AW = advanced women; AM = advanced men; * = *p* < 0.05 vs. AW; # = *p* < 0.05 vs. NW.

**Table 3 ijerph-21-00891-t003:** The nutrition profile of cross-training practitioners.

Variables	NW (*n* = 17)	NM (*n* = 15)	AW (*n* = 17)	AM (*n* = 16)	X^2^	*p*-Value
Energy supplements	10 (58.8%)	9 (60.0%)	11 (64.7%)	12 (75.0%)	1.14	0.7
Vitamin supplements	6 (35.3%)	3 (20.0%)	6 (35.3%)	7 (43.8%)	2.01	0.5
Anabolic steroids	1 (5.9%)	1 (6.7%)	4 (23.5%)	3 (18.8%)	3.21	0.3

Note: data are presented in frequency and percentage for only positive reports of consumption supplements and ergogenic factors. *n* = number of participants; NW = novice women; NM = novice men; AW = advanced women; AM = advanced men.

**Table 4 ijerph-21-00891-t004:** RMR measured by IC, BIA, and predictive equations of cross-training practitioners.

Groups	NW (*n* = 17)	NM (*n* = 15)	AW (*n* = 17)	AM (*n* = 16)
BIA	1362.5 (91.2)	1935.3 (150.0) #	1455.0 (160.6)	1936.0 (208.9) *
IC	1274.6 (209.5)	2147.3 (319.9) #	1529.6 (375.7)	2069.4 (469.8) *
Harris–Benedict	1533.4 (111.7)	1984.4 (152.0) #	1426.2 (84.1)	1929.4 (217.7) *
Cunningham	1059.0 (55.6)	1414.9 (87.7) #	1117.8 (98.4)	1421.1 (130.9) *
De Lorenzo	1618.0 (103.0)	2049.9 (141.5) #	1636.0 (157.6)	1965.3 (171.0) *
Tinsley (a)	1525.7 (140.9)	2205.2 (271.1) #	1587.5 (220.5)	2138.1 (327.2) *
Tinsley (b)	942.1 (65.4)	2205.9 (271.1) #	1011.3 (115.9)	1368.4 (154.1) *
Johnstone	962.0 (65.78)	1314.0 (105.1) #	1004.0 (95.87)	1305.0 (158.7) *

Note: data are presented as mean ± standard deviation; *n* = number of participants; NW = novice women; NM = novice men; AW = advanced women; AM = advanced men; IC = indirect calorimetry; * = *p* < 0.05 vs. AW; # = *p* < 0.05 vs. NW.

**Table 5 ijerph-21-00891-t005:** Resting metabolic rate responses adjusted for the body mass of cross-training groups.

Groups	CW (*n* = 34)	CM (*n* = 31)
IC	1568.5 (1513.7–1623.3)	1868.8 (1826.8–1910.8)
BIA	1577.1 (1523.1–1632.1)	1722.7 (1681.3–1764.1) #
Harris–Benedict	1654.6 (1600.6–1708.7)	1742.3 (1700.8–1783.7) #
Cunningham	1195.1 (1141.0–1249.1) *	1283.0 (1241.6–1324.4) #
De Lorenzo	1833.7 (1779.6–1887.7) *	1780.6 (1739.2–1822.0) #
Tinsley (a)	1759.8 (1705.7–1813.8) *	1934.7 (1893.3–1976.1)
Tinsley (b)	1065.8 (1011.7–1119.8) *	1240.4 (1199.0–1281.8) #
Johnstone	1072.4 (051.8–1171.4) *	1195.6 (1119.8–1202.7) #

Note: data are presented as mean and 95% CI = confidence interval; n = number of participants; CW = CT women; CM = CT men; IC = indirect calorimetry; * = *p* < 0.05 vs. IC in CW; # = *p* < 0.05 vs. IC in CM.

**Table 6 ijerph-21-00891-t006:** Resting metabolic rate adjusted for the skeletal muscle mass of cross-training groups.

Groups	CW (*n* = 34)	CM (*n* = 31)
IC	1649.8 (1598.5–1710.2)	1805.6 (1763.5–1847.6)
BIA	1651.3 (1591.5–1711.1)	1668.1 (1626.6–1709.6) #
Harris–Benedict	1724.2 (1664.5–1784.0)	1683.0 (1641.5–1724.4) #
Cunningham	1244.1 (1184.4–1303.9) *	1247.0 (1205.6–1288.5) #
De Lorenzo	1918.0 (1858.2–1977.8) *	1720.6 (1679.1–1762.0) #
Tinsley (a)	1838.7 (1778.9–1898.4) *	1864.0 (1822.5–1905.5)
Tinsley (b)	1107.9 (1048.1–1167.7) *	1206.2 (1164.8–1247.7) #
Johnstone	1111.6 (1051.8–1171.4) *	1161.2 (1119.8–1202.7) #

Note: data are presented as mean and 95% CI = confidence interval; *n* = number of participants; CW = CT women; CM = CT men; IC = indirect calorimetry; * = *p* < 0.05 vs. IC in CW; # = *p* < 0.05 vs. IC in CM.

## Data Availability

The original contributions presented in the study are included in the article, further inquiries can be directed to the corresponding author.

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
