# Peer review of "Comparison between Measured and Predicted Resting Metabolic Rate Equations in Cross-Training Practitioners"

_ijerph, 2024, doi:10.3390/ijerph21070891_

Round 1

Reviewer 1 Report

Comments and Suggestions for Authors

Dear Authors,

The study addresses a significant issue in sports science, the accurate measurement of RMR, which is critical for optimizing training and nutrition strategies for CT practitioners. The methods are well-detailed, including a thorough assessment of anthropometric and body composition characteristics, RMR measurements, and nutritional profiles. The use of multiple statistical methods, including Bland-Altman plots, provides a good understanding of the agreement between IC and predictive equations. The discussion effectively translates the findings into practical recommendations for professionals in sports nutrition and training. However, there are some points that need addresing and further explanations. Find those in the following comments below, separate for each section of the paper:

Introduction: The introduction effectively sets the stage by explaining what CT entails, including the variety of exercises and their benefits. While the introduction contains pertinent information, it can benefit from improved clarity and flow. The transition between discussing CT and the importance of RMR could be smoother. Also, the review of existing literature on RMR and predictive equations could be more detailed. Including more specific examples of studies that highlight the discrepancies between measured and predicted RMR in various populations would strengthen the background. The objective of the study is mentioned at the end, but it could be stated more prominently earlier in the introduction to provide readers with a clear understanding of the study's aim from the beggining. Further, some technical terms (e.g., "respiratory quotient") are well-defined, which is good, but more explanation could be given for non-specialist readers. So, please, reorganize the introduction to ensure a smoother transition between discussing CT and the importance of RMR estimation. Consider breaking down long sentences to improve readability. 

Methods: The study employs a cross-sectional and comparative design, which is appropriate for examining differences between groups of CT practitioners. The rationale for the sample size is grounded in prior research, enhancing the study's credibility. The paper could benefit from a more detailed explanation of how participants were randomized to avoid selection bias. More information on how confounding variables (e.g., dietary intake, sleep patterns) were controlled would be valuable. Also, additional demographic information (e.g., ethnicity, socioeconomic status) could provide more context and help understand the generalizability of the findings. Further, describing the conditions under which measurements were taken (e.g., time of day, fasting state) would enhance reproducibility. And discussing potential limitations or sources of error in the use of indirect calorimetry would provide a more balanced view. Regarding data processing, reporting effect sizes alongside p-values would provide more information about the practical significance of the findings.

Discussion: The discussion section effectively summarizes the key findings of the study and places them in the context of existing research. The authors provide a clear narrative linking the results to practical implications for CT practitioners. However, the section could be improved in clarity, organization, and depth in interpreting the results. Firstly, explaining why certain equations might underestimate or overestimate RMR, considering factors like body composition variables, could provide more insight. Additionally, discussing the implications of these findings for practitioners in terms of how they might adjust training or dietary regimens would add practical value. The finding that there are no significant differences in the consumption of energy supplements, vitamins, or anabolic steroids among the groups is interesting, especially in contrast to previous studies. The authors could expand on this by hypothesizing why CT practitioners might have a homogeneous nutritional profile, possibly due to similar training environments or nutritional education. The discussion appropriately highlights the importance of body mass in the accuracy of RMR measurements. However, a more straightforward presentation of which methods are most reliable for each subgroup (CW and CM) is needed. Maybe summarizing these points in a table within the discussion might help clarify the practical applications of these findings (think about this, but not obligatory). The limitations of the study are acknowledged, including the use of IC as the gold standard and the study's focus on competitive practitioners. The authors might consider suggesting future research directions to address these limitations, such as studies involving different levels of training periodization or non-competitive practitioners. 

Conclusion: This sections was written pretty modest. Consider adding some practical implications here (maybe in a manner of bulletpoints) with most interesting findings from the study.

Author Response

The study addresses a significant issue in sports science, the accurate measurement of RMR, which is critical for optimizing training and nutrition strategies for CT practitioners. The methods are well-detailed, including a thorough assessment of anthropometric and body composition characteristics, RMR measurements, and nutritional profiles. The use of multiple statistical methods, including Bland-Altman plots, provides a good understanding of the agreement between IC and predictive equations. The discussion effectively translates the findings into practical recommendations for professionals in sports nutrition and training. However, there are some points that need addresing and further explanations. Find those in the following comments below, separate for each section of the paper:

Introduction: The introduction effectively sets the stage by explaining what CT entails, including the variety of exercises and their benefits. While the introduction contains pertinent information, it can benefit from improved clarity and flow. The transition between discussing CT and the importance of RMR could be smoother. Also, the review of existing literature on RMR and predictive equations could be more detailed. Including more specific examples of studies that highlight the discrepancies between measured and predicted RMR in various populations would strengthen the background.

Our reply

Thank you for your note. We have improved the transition between CT and RMR (lines 58-62). Regarding the inclusion of more specific examples, we improved the introduction by highlighting the discrepancies between the measured and predicted RMR, strengthening the context as requested by the reviewer (lines 78-94).

The objective of the study is mentioned at the end, but it could be stated more prominently earlier in the introduction to provide readers with a clear understanding of the study's aim from the beggining. Further, some technical terms (e.g., "respiratory quotient") are well-defined, which is good, but more explanation could be given for non-specialist readers.

Our reply

Thank you for your note. We have improved the explanation in lines 69-70 as suggested by the reviewer.

So, please, reorganize the introduction to ensure a smoother transition between discussing CT and the importance of RMR estimation. Consider breaking down long sentences to improve readability. 

Our reply

We appreciate your suggestion and breaking down long sentences to improve readability, as requested by the reviewer.

Methods: The study employs a cross-sectional and comparative design, which is appropriate for examining differences between groups of CT practitioners. The rationale for the sample size is grounded in prior research, enhancing the study's credibility. The paper could benefit from a more detailed explanation of how participants were randomized to avoid selection bias. More information on how confounding variables (e.g., dietary intake, sleep patterns) were controlled would be valuable. Also, additional demographic information (e.g., ethnicity, socioeconomic status) could provide more context and help understand the generalizability of the findings. Further, describing the conditions under which measurements were taken (e.g., time of day, fasting state) would enhance reproducibility. And discussing potential limitations or sources of error in the use of indirect calorimetry would provide a more balanced view.

Our reply

Thank you for your note. The allocation of participants was intentional according to the following criteria:( i) practice CT regularly for at least 6 months (novice and advanced); (ii) advanced practitioners self-identify as one and have competed in the last two years; and (iii) training frequency > 3 sessions a week for participants of the two groups practitioners (novice and advanced). We have improved the description of the intentional allocation of participants in section 2.2. In addition, we added the assessment time, location and city and improved the description of the instructions for the assessments in section 2.1. Regarding the confounding variables, we assumed that this was a limitation of the study, considering that we did not collect a dietary record and information regarding sleep patterns. These limitations were described at the end of the discussion session. Finally, we improve the description of bioelectrical impedance-BIA and measurement of RMR via Indirect Calorimetry in section 2.3 and 2.4 respectively.

Regarding data processing, reporting effect sizes alongside p-values would provide more information about the practical significance of the findings.

Our reply

Thank you for your suggestion. The effect size was determined using Cohen’s d, categorized as follows: small effect (0.2), moderate effect (0.5) and large effect (0.8), considering the values adjusted by ANCONVA.

Formulas used for the calculation:

On what:

M1 and M2 are the adjusted means of the two groups.

SD1 and SD2 are the adjusted standard deviations of the two groups.

n1 and n2 are the sample sizes of the two groups.

Finally, we report p-values as suggested by the reviewer.

iscussion: The discussion section effectively summarizes the key findings of the study and places them in the context of existing research. The authors provide a clear narrative linking the results to practical implications for CT practitioners. However, the section could be improved in clarity, organization, and depth in interpreting the results. Firstly, explaining why certain equations might underestimate or overestimate RMR, considering factors like body composition variables, could provide more insight. Additionally, discussing the implications of these findings for practitioners in terms of how they might adjust training or dietary regimens would add practical value.

Our reply

Thank you for your note. We have improved the discussion in terms of clarity, organization, and depth in interpreting the results, and we have discussed the practical implications of the findings as requested by the reviewer.

The finding that there are no significant differences in the consumption of energy supplements, vitamins, or anabolic steroids among the groups is interesting, especially in contrast to previous studies. The authors could expand on this by hypothesizing why CT practitioners might have a homogeneous nutritional profile, possibly due to similar training environments or nutritional education.

Our reply

Thank you for your observation. Our sample is homogeneous; therefore, it is not possible to extrapolate the present findings to populations with different conditions. We inserted this information into the discussion as suggested by the reviewer.

The discussion appropriately highlights the importance of body mass in the accuracy of RMR measurements. However, a more straightforward presentation of which methods are most reliable for each subgroup (CW and CM) is needed. Maybe summarizing these points in a table within the discussion might help clarify the practical applications of these findings (think about this, but not obligatory).

Our reply

Thank you for your note. We reformulated the first paragraph of the discussion and made an effort to directly present the most reliable methods for each group as requested by the reviewer.

The limitations of the study are acknowledged, including the use of IC as the gold standard and the study's focus on competitive practitioners. The authors might consider suggesting future research directions to address these limitations, such as studies involving different levels of training periodization or non-competitive practitioners. 

Our reply

Thank you for the suggestion. We have included a paragraph on future research at the end of the discussion, as recommended by the reviewer.

Conclusion: This sections was written pretty modest. Consider adding some practical implications here (maybe in a manner of bulletpoints) with most interesting findings from the study.

Our reply

Thank you for your note. We made an effort to improve the conclusion as suggested by reviewer.

Reviewer 2 Report

Comments and Suggestions for Authors

Dear Authors,

I am pleased you have attempted to address the problem of studying resting metabolic rate (RMR) in cross-trained individuals (advanced and beginners).

In my opinion, your line of research is important from the point of view of preparation for sports competition.

It is all the more important in the Olympic year.

Methodological evaluation:

The survey was conducted on 66 people.

It seems insufficient - so please report the results to G*Power.

If the result is unsatisfactory, I suggest (to make the study more solid) to add in the title, for example - the first stage of the study, or - a pilot study.

The logical consequence of this will be without studying a larger population in other projects.

The choice of statistical methods is appropriate. I have no objections to this.

Substantive evaluation:

The authors cited 44 items of literature in their study, which is one of many conducted worldwide in this area.

I suggest that the number of global references should be increased, this will raise the merit of the problem under consideration.

Consider here on the issue of “Body Composition” - Witkowski K et al. Body composition and motor potential of judo athletes in selected weight categories. Arch Budo 2021; 17: 161-175 

The conclusions are correctly derived.

Technical evaluation:

The abstract lacked background. Complete it.

The “Introduction” should be expanded. The current form is somewhat superficial.

At the end of the discussion, add subsections “Limitations of the study” and “Directions for further research”. - This is important, assuming follow-up.

Conclusion:

The article can contribute to physical culture sciences in the field of sports.

But the authors must make the indicated corrections in order for the article to proceed further.

Author Response

I am pleased you have attempted to address the problem of studying resting metabolic rate (RMR) in cross-trained individuals (advanced and beginners). In my opinion, your line of research is important from the point of view of preparation for sports competition. It is all the more important in the Olympic year.

Methodological evaluation: The survey was conducted on 66 people. It seems insufficient - so please report the results to G*Power. If the result is unsatisfactory, I suggest (to make the study more solid) to add in the title, for example - the first stage of the study, or - a pilot study. The logical consequence of this will be without studying a larger population in other projects. The choice of statistical methods is appropriate. I have no objections to this.

Our reply

A probabilistic sample based on a previous study with the same methodology indicated that 61 participants were enough to identify an α = 0.05 and β = 0.80. Our sample contains 65 participants. Therefore, according to the sample calculation presented, we believe that the sample is sufficient. You can find this information in the section 2.1.

Substantive evaluation: The authors cited 44 items of literature in their study, which is one of many conducted worldwide in this area. I suggest that the number of global references should be increased, this will raise the merit of the problem under consideration. Consider here on the issue of “Body Composition” - Witkowski K et al. Body composition and motor potential of judo athletes in selected weight categories. Arch Budo 2021; 17: 161-175 

Our reply

Thank you for the reference suggestion. We used it to discuss the findings is the lines 438-440 as suggested by reviewer.  

The conclusions are correctly derived.

Our reply

Thank you for your note. No response is needed.

Technical evaluation: The abstract lacked background. Complete it.

Our reply

Thank you for your note. We completed the abstract. 

The “Introduction” should be expanded. The current form is somewhat superficial.

Our reply

Thank you for your suggestion. We made an effort to expand the introduction as suggested by reviewer.

At the end of the discussion, add subsections “Limitations of the study” and “Directions for further research”. - This is important, assuming follow-up.

Thank you for your suggestion. We improved the study's limitations, practical implications, and suggested future research as recommended by the reviewer

Conclusion:

The article can contribute to physical culture sciences in the field of sports. But the authors must make the indicated corrections in order for the article to proceed further.

Reviewer 3 Report

Comments and Suggestions for Authors

Many thanks for the opportunity to review the scientific paper "Comparison between measured and predicted resting meta- 2 bolic rate equations in Cross Training Practitioners" 

The high standard is evident when the entire scientific article was analysed. The authors used many analyses of variance in the paper, e.g. ANOVA or ANCOVA covariance. The authors also used the Bonferroni test. The multitude of tests used demonstrates the high quality of the paper.  

In the material and methods section, the authors made a fair division into four groups: women and men (novices and professionals). In the future, it is recommended to increase the study group. The authors describe how the selection of the above mentioned division was made.  

In the section: introduction - suggests not to use the words ‘the best nutritional strategy’ - if it is the best strategy then please give a citation.

Please explain under the tables the abbreviation: IC - no explanation (table 4, 5, 6). 
Line: 394 I propose make a correct ‘which is most’ to ‘which is the most’
Line: 399 the word with is written in cursive. 
In the Literature section, please check or add missing pages in the following citations: 16, 20, 36, 40, 41, 43. 

Suggestion whether citation number 38 (Fukagawa from 1980 should not be changed to a scientific publication from the last 20 years) - if not please explain why a publication from 54 years ago is worth citing.

Author Response

Many thanks for the opportunity to review the scientific paper "Comparison between measured and predicted resting meta- 2 bolic rate equations in Cross Training Practitioners" 
The high standard is evident when the entire scientific article was analysed. The authors used many analyses of variance in the paper, e.g. ANOVA or ANCOVA covariance. The authors also used the Bonferroni test. The multitude of tests used demonstrates the high quality of the paper.   In the material and methods section, the authors made a fair division into four groups: women and men (novices and professionals). In the future, it is recommended to increase the study group. The authors describe how the selection of the above mentioned division was made.  

In the section: introduction - suggests not to use the words ‘the best nutritional strategy’ - if it is the best strategy then please give a citation.

Our reply

Thank you. We correct the expression.

Please explain under the tables the abbreviation: IC - no explanation (table 4, 5, 6). 

Our reply

Thank you for your correction.

Line: 394 I propose make a correct ‘which is most’ to ‘which is the most’

Our reply

Thank you for your correction.

Line: 399 the word with is written in cursive. 

Our reply

Thank you for your correction.

In the Literature section, please check or add missing pages in the following citations: 16, 20, 36, 40, 41, 43. 

Our reply

Thank you for your correction. We have added the pages to the indicated references.

Suggestion whether citation number 38 (Fukagawa from 1980 should not be changed to a scientific publication from the last 20 years) - if not please explain why a publication from 54 years ago is worth citing.

Our reply

Thank you for the observation. Indeed, it is an old publication. Therefore, to improve and strengthen our arguments, we used: Tinsley GM, Graybeal AJ, Moore ML. Resting metabolic rate in muscular physique athletes: validity of existing methods and development of new prediction equations. Appl Physiol Nutr Me 2018, 44, 397-406.

Round 2

Reviewer 2 Report

Comments and Suggestions for Authors

Dear Authors,

Thank you very much for your detailed replies to the review.

I am glad that the article was corrected and the changes were marked in a different yellow color, which made it easier for me to identify the new quality.

Now the article meets the journal's criteria and I recommend it for publication.